# Bmi1 Augments Proliferation and Survival of Cortical Bone-Derived Stem Cells after Injury through Novel Epigenetic Signaling via Histone 3 Regulation

**DOI:** 10.3390/ijms22157813

**Published:** 2021-07-22

**Authors:** Lindsay Kraus, Chris Bryan, Marcus Wagner, Tabito Kino, Melissa Gunchenko, Wassy Jalal, Mohsin Khan, Sadia Mohsin

**Affiliations:** 1Independence Blue Cross Cardiovascular Research Center, Lewis Katz School of Medicine at Temple University, Philadelphia, PA 19140, USA; tuj16109@temple.edu (L.K.); cfbryan1447@gmail.com (C.B.); wagner@temple.edu (M.W.); tabito.kino@temple.edu (T.K.); melissa.gunchenko@temple.edu (M.G.); wizjalal@temple.edu (W.J.); 2Center for Metabolic Diseases, Lewis Katz School of Medicine at Temple University, Philadelphia, PA 19140, USA; mohsin.khan@temple.edu

**Keywords:** stem cells, proliferation, survival and cardiac repair

## Abstract

Ischemic heart disease can lead to myocardial infarction (MI), a major cause of morbidity and mortality worldwide. Multiple stem cell types have been safely transferred into failing human hearts, but the overall clinical cardiovascular benefits have been modest. Therefore, there is a dire need to understand the basic biology of stem cells to enhance therapeutic effects. Bmi1 is part of the polycomb repressive complex 1 (PRC1) that is involved in different processes including proliferation, survival and differentiation of stem cells. We isolated cortical bones stem cells (CBSCs) from bone stroma, and they express significantly high levels of Bmi1 compared to mesenchymal stem cells (MSCs) and cardiac-derived stem cells (CDCs). Using lentiviral transduction, Bmi1 was knocked down in the CBSCs to determine the effect of loss of Bmi1 on proliferation and survival potential with or without Bmi1 in CBSCs. Our data show that with the loss of Bmi1, there is a decrease in CBSC ability to proliferate and survive during stress. This loss of functionality is attributed to changes in histone modification, specifically histone 3 lysine 27 (H3K27). Without the proper epigenetic regulation, due to the loss of the polycomb protein in CBSCs, there is a significant decrease in cell cycle proteins, including Cyclin B, E2F, and WEE as well as an increase in DNA damage genes, including ataxia-telangiectasia mutated (ATM) and ATM and Rad3-related (ATR). In conclusion, in the absence of Bmi1, CBSCs lose their proliferative potential, have increased DNA damage and apoptosis, and more cell cycle arrest due to changes in epigenetic modifications. Consequently, Bmi1 plays a critical role in stem cell proliferation and survival through cell cycle regulation, specifically in the CBSCs. This regulation is associated with the histone modification and regulation of Bmi1, therefore indicating a novel mechanism of Bmi1 and the epigenetic regulation of stem cells.

## 1. Introduction

Worldwide, there are approximately 32.4 million myocardial infarctions (MIs) every year [1]. Damage to the heart after MI is often irreversible [2,3]; however, stem cell therapy has been studied as a potential therapeutic intervention after cardiac injury [4,5]. Stem cells have unique characteristics that enable growth, regeneration, and cardiac healing after insult [6,7]. Each stem cell type used in cardiac wound healing has different attributes that have been suggested for reparative function [7,8]. However, to date, all stem cell clinical trials in the heart have shown modest improvement, emphasizing the need to better understand the basic biology of the cells and processes that stem cells target after transplantation [9,10]. Recently, cortical bone stem cells (CBSCs) have been shown to have cardioprotective effects in small and large animal heart failure models [11,12,13,14]. The underlying mechanism of how CBSCs mediate reparative effects after injury is still unknown. CBSCs have shown increased paracrine signaling, which is a major attribute for stem cell therapies. Paracrine signaling in stem cells is often attributed to factors that promote cardioprotection [14], specifically through increased proliferation, decreased cell death, and increased cell cycle regulation [15]. These beneficial cellular mechanisms are strongly regulated by epigenetics [16,17].

The polycomb repressive complexes (PRC1 and PRC2) epigenetic regulation has been strongly associated with stem cell cycle regulation, proliferation, and cell death [18,19]. The PRC1 is composed of the main epigenetic regulator Bmi1, as well as chromobox protein (CBX), RYBP, and RING [20]. Bmi1 is a critical regulator for cardiac reprogramming through the ubiquitination of histone 2A on lysine 119 [21]. Concurrently, there are several studies that have highlighted the importance of Bmi1 in different types of stem cells. There have been several reports that link Bmi1 epigenetic regulation to cell cycle regulation [22,23], specifically in adult murine hematopoietic stem cells (HSCs) [24], neural stem cells (NSCs) [25], and cardiac progenitor cells (CPCs) [26]. There has been a report of stem cell death in ischemic environments, which makes the role of Bmi1 important to understand because of the potential protective and proliferative role under stress [27]. Bmi1 and polycomb proteins have been strongly connected to the INK/ARF pathway of cell proliferation and survival [28]. Specifically, Bmi1 has been associated with beneficial DNA repair mechanisms through epigenetic regulation, increasing ATM and ATR response to double strand break in DNA [29].

Stem cells have promising potential; however, current stem cell therapies have not been as successful as originally hoped. This is often due to a loss of engraftment, pluripotency, and differentiation [30]. The recently discovered CBSCs have been shown to overcome these obstacles [13]; however, the mechanism behind CBSCs was previously unknown. CBSCs have shown enhanced cardiac wound healing and increased heart function following injury, which was attributed to unique paracrine signaling [12]. Nevertheless, the specific CBSC mechanisms that provide this therapeutic potential was not known. This article provides insight into the CBSC mechanism by the epigenetic regulation of the PRC1 protein, Bmi1.

## 2. Results

### 2.1. Expression of Bmi1 in CBSCs

CBSCs have been shown to have unique functions compared to other stem cell types; however, until now, the detailed epigenetic differences were not as well understood. We show that there is an increase in Bmi1 expression in CBSCs compared to the bone marrow-derived mesenchymal stem cells (MSCs) as well as cardiac-derived stem cells (CDCs) in both protein (Figure 1A) and RNA (Figure 1B) using Western blotting and rt-PCR analysis, respectively (*p* value 0.026 and 0.011 for Western blotting and *p* value 0.0003 and 0.031 for PCR). When CBSCs are exposed to hypoxic conditions (600 μM hydrogen peroxide), there is an increase in Bmi1 expression compared to normal CBSC conditions (Figure 1C) (*p* value = 0.042), indicating the importance of Bmi1 in a stressed and altered state. Additionally, genes vital for cell cycle regulation and cell survival are essential in CBSCs compared to the other stem cells, CDC and MSCs (Figure 1D). So initially, CBSCs have increased Bmi1 expression as well as changes in cell cycle regulation and cell survival genes compared to other stem cell types. These changes are the basis for the important differences seen in CBSCs.

### 2.2. Bmi1 Regulates the Proliferative Potential of CBSCs

Bmi1 was significantly knocked down using the lentivirus, as shown in Figure 2A by immunoblot analysis. Loss of Bmi1 in CBSCs leads to a significant decrease in proliferative ability versus control cells as seen by representative images in Figure 2B. There is a significant decrease in cell number of Bmi1 knockdown (KD) CBSCs in culture after 4, 5, and 6 days (*p* < 0.0001 for last 3 days) (Figure 2C), indicating the importance of Bmi1 in cell proliferation. The decrease in the proliferative capacity was further confirmed by CyQUANT assay. The Bmi1 KD showed a significant decrease in proliferation compared to control cells after 3, 4, and 5 days (*p* = 0.0002, <0.0001, <0.0001, respectively) (Figure 2D) indicating a decrease in cellular proliferation associated with a decrease in DNA content. Using the MTT assay to assess proliferation and metabolic activity in the Bmi1 KD CBSCs compared to the control, there is a significant decrease in proliferation after 4 days (*p* < 0.0001) (Figure 2E), indicating a decrease in metabolic activity without Bmi1.

### 2.3. Bmi1 Knockdown Increases Apoptosis and Causes DNA Damage in CBSCs

To assess DNA damage, comet assays were used in the Bmi1 KD and Control CBSCs as shown in the representative images in Figure 3A. When Bmi1 is knocked down, there are larger comets as measured by the percent of DNA in the tail of the comet (*p* < 0.0001), the length of the comet (*p* < 0.0001), and the olive moment (tail versus the head of the comet) (*p* < 0.0001). All comet analyses indicate more DNA damage with a loss of Bmi1 in CBSCs (Figure 3B). Additionally, there is an increased percentage of CBSCs positive for Annexin V and DAPI without Bmi1, measured by FACS analysis, (Figure 3C), which indicates increased apoptosis and necrosis, respectively, compared to control cells. Without Bmi1, 67.9% of CBSCs are double positive for Annexin V and DAPI versus control CBSCs which only have 10.7% double positive population (Figure 3C), demonstrating a substantial increase in cell death without Bmi1. Expression of DNA damage response genes, including ataxia-telangiectasia mutated (ATM) and ATM and Rad3-related (ATR), were significantly decreased in the Bmi1 KD (Figure 3D), indicating a loss of regulated DNA damage repair without Bmi1, as measured by RT-PCR analysis.

### 2.4. Bmi1 Knockdown Causes Cell Cycle Arrest in CBSCs

In Bmi1 KD CBSCs, there are significant changes in the cell cycle progression, which result in the cell cycle arrest of CBSCs compared to control CBSCs. Representative images of the cell sorting for the cell cycle progression are shown in Figure 4A. There are significantly less Bmi1 KD CBSCs in the G1 stage (*p* value 0.007); however, there are significantly more in the G2 phase of the cell cycle (*p* value 0.003) (Figure 4B), signifying a dysregulation of cell cycle progression of CBSCs without Bmi1. Additionally, using RT-PCR analysis (Figure 4C), there is a significant increase in the cell cycle inhibitor WEE (*p* value 0.001), as well as the tumor suppressor gene, E2F (*p* value 0.023), both associated with cell cycle regulation and cell cycle arrest. Interestingly, there is a decrease in the cyclin B (CCNB1) (*p* value 0.041), which is directly connected to the cell cycle transition of the G2 to M phase.

### 2.5. Bmi1 Modifies Epigenetic Regulation in CBSCs

Using Western blot analysis, variations of histone 3 were analyzed, including histone 3, trimethylated H3, and acetylated histone 3. Without Bmi1 in CBSCs, there was a significant decrease in H3 (*p* value 0.006) and trimethylated histone 3 (*p* value of 0.050), with no changes seen in acetylated histone 3 (Figure 5A), demonstrating the significant epigenetic changes associated with Bmi1 in CBSCs. Due to the changes seen in Figure 5A, it was necessary to further explore the epigenetic regulation through ModSpec analysis to better understand the epigenetic mechanisms associated with Bmi1 in CBSCs. Specifically, there were significant differences in the percent of histone modifications, specifically methylation at histone 3 lysine 27 (H3K27) (Figure 5B). Finally, a schematic of the mechanism of Bmi1 in CBSC function diagrams the importance of the H3K27 modification on CBSC proliferation and survival as shown in Figure 5C. The loss of Bmi1 in CBSCs results in a decrease in histone modification at histone 3 lysine 27. The loss of this epigenetic modification is associated with a decrease in cell survival through an apoptotic mechanism associated with DNA repair. Additionally, there is a decrease in cellular proliferation regulated through the cell cycle modulation by WEE, E2F, and Cyclin B in CBSCs with a loss of Bmi1 (Figure 5C).

## 3. Discussion

Our findings identify a novel mechanism of cortical bone stem cells (CBSCs) and their therapeutic potential, specifically through the epigenetic regulation of Bmi1. In particular, CBSCs overexpress Bmi1 compared to other stem cells types, which enhanced the regulation of cellular proliferation, DNA damage, and cell cycle. Without Bmi1, these important cellular mechanisms are decreased, resulting in attenuated CBSCs, as depicted in Figure 5C.

Interestingly, without Bmi1 in both mouse and human stem cells, there is also an increase in cell death and apoptosis [24]. Bmi1 is necessary to regulate cell proliferation normally; however, when there is stress or injury, the Bmi1 regulation is lost [31]. For example, the overexpression of Bmi1 has been connected to decrease in oxidative stress associated with reactive oxygen species (ROS), which allowed for the regeneration of hematopoietic stem cells [31], making it an important target for regulating repair after cardiac injury.

Through epigenetic regulation by Bmi1, CBSCs maintain proliferation, regulate survival, and control the cell cycle. Bmi1 has been well studied in regard to these cellular mechanisms, specifically in cancer cells [32]. In these studies, Bmi1 is closely associated with the INK4a/ARF locus of cell cycle regulation in adult and neonatal models [33,34,35]. Without polycomb proteins like Bmi1, there is disrupted development, cell cycle progression, and organ growth, indicating its necessity in cellular function in the INK4a/ARF pathway [35]. However, the role of Bmi1 in stem cells has been lacking, despite the therapeutic connection to stem cell research. Studies have shown that stem cells positive for Bmi1 have increased reparative function [36]. Additionally, Bmi1 has proven vital for the maintenance stem cell properties [24]. Our results show that without the Bmi1, there is a loss of the epigenetic regulation resulting in dysregulated marks at histone 3 lysine 27 (H3K27). The H3K27 modification was associated with gene silencing as a repressive marker, specifically by polycomb proteins [37]. Moreover, the H3K27 mark is an abundant modification that has multiple variations that can contribute to the silencing of the gene expression relative to the proliferation and survival mechanisms [38]. Previously, without this H3K27 mark as well as the H2AK119, there is a lack of chromatin silencing which leads to tumor-like cell growth in somatic cells [28,39]. However, this research is indicating the importance of the regulation of the H3K27 modification associated with the polycomb complex in CBSC function. The trimethylation modification at histone 3 is decreased without Bmi1 which also causes an attenuation of CBSC proliferation. Because of the loss of Bmi1 and therefore the enhanced repression through the H3K27 epigenetic modification, there is a decrease in proliferation, increase in DNA damage, and the dysregulation of the cell cycle. A key feature of stem cells, and specifically CBSCs, has been their unique paracrine signaling and wound healing properties, which we are suggesting is due to the increase in the epigenetic regulation by Bmi1. The CBSCs have a higher expression of Bmi1 which indicates enhanced epigenetic regulation, allowing for increased proliferative potentiation and survival, demonstrating the therapeutic mechanisms of these novel stem cells.

Understanding the epigenetic regulation of stem cells is a promising therapeutic strategy for disease diagnosis and treatment. Particularly after myocardial injury, CBSC therapy has been shown to significantly improve heart structure and function. Prior to this study, the mechanism of CBSCs in the failing heart was unknown. Presently, we showed that the loss of Bmi1 in CBSCs results in a loss of histone 3 epigenetic modifications; therefore, reducing cell survival and proliferation of CBSCs (Figure 5C). This study provides a novel role and possible beneficial mechanism of Bmi1 in CBSCs. Moreover, targeting Bmi1 in a clinical setting, specifically with stem cell therapy, could be an instrumental target to enhance and maintain CBSC proliferation and survival in a failing heart.

The clinical implication of this study provides a deeper understanding and potential target of CBSC stem cell therapy in heart failure. Epigenetic biomarkers are diverse; therefore, there are a multitude of variations on the function and downstream effects of epigenetic modifications in human diseases. More research is needed on the downstream effects of epigenetic modifications in stem cell therapy. Furthermore, it would be vital to study the role of the PRC2 and PRC1 together in CBSCs as the polycomb complexes are similar in function. Additionally, in future studies, assessing the role of the overexpressing Bmi1 in an injury condition with the CBSCs would provide an important assessment of the functionality of the stem cell and its mechanism. Currently, we are reporting a novel role of epigenetic regulation in the promising cortical bone stem cells, thereby enhancing the understanding of epigenetic regulation of stem cell biology.

## 4. Materials and Methods

### 4.1. Cell Culture and Bmi1 knockdown

Cortical bone-derived stem cell culture: cortical bone-derived stem cells were isolated from the tibias and femurs of C57BL/6 mice, as previously described [13,21]. Briefly, tibias and femurs were flushed to remove all the bone marrow and then digested in collagenase at 37 °C. The digested cells were washed and plated in CBSC media until colonies of CBSCs appeared. The cells were characterized for CBSC markers and expanded for experiments.

Mesenchymal stem cell culture: the femur and tibia of C57BL/6J mice ranging from 2 to 6 months of age were isolated. The epiphysis of each were removed and each bone cavity was flushed with pre-warmed, and the complete culture medium for bone marrow extracted. Extracted marrow medium was passed through a 70 μM filter and remaining cell suspension was cultured in DMEM (Corning) containing 20% fetal bovine serum (Neuromics, Edina, Min, USA) and 1% penicillin streptomycin/L-glutamine (Gibco Life Technologies, Waltham, MA, USA). MSCs were isolated from other bone marrow cells via two passages of plastic adherence. Remaining cells were expanded and cryopreserved as previously described [40].

Cardiac-derived stem cells culture: cardiac-derived stem cells RNA and protein were generously gifted to us from Dr. Mohsin Khan’s laboratory.

### 4.2. Heat Map Comparison

A comparison between CBSC, CDC, and MSC genes for cell cycle and survival was constructed using RNA sequencing as previously described [40]. Raw fastq files were indexed to mouse genome mm10 and quantified for downstream analysis by Salmon. The bioinformatic analysis was performed using quantified transcripts imported as a matrix of average transcript length using package tximport. to package DESeq2 for downstream differential expression analysis in R. Genes were identified and those with less than 5 reads per samples were removed. Wald test was used to statistically identify genes with *p* value < 0.05. Using regularized logarithm (rlog) function, count data were transformed for visualization on a log2 scale.

### 4.3. Lentivirus Transduction

Using the CBSCs in culture, the lentivirus from Vector Labs for Bmi1 and the control “Scramble” were used following Vector Labs instructions. Both lentiviruses are an shRNA knockdown vector. Briefly, in 6-well plates of 50,000 CBSCs per well, an MOI of 10 was used for each virus, which was determined using the given titer from Vector Labs. The lentivirus was added in 1 mL of CBSC media and let set overnight. The next day, the wells were washed and CBSC media were added. After 72 h, the CBSCs were used for experimentation.

### 4.4. RT-PCR and Western Blotting

Quantitative real-time PCR and RT2 Profiler PCR Arrays: CBSCs were tested for expression of proliferation, DNA damage, cell cycle, and PRC1 genes by using RT2 profiler PCR arrays (Qiagen, Hilden, Germany). Briefly, RNA was isolated from cells using the miRNeasy Kit (Qiagen) according to the manufacturer’s protocol. Single-stranded cDNA was synthesized from all samples using the RT2 First Strand Kit (Qiagen) as described in the Qiagen protocol for RT2 profiler array sample preparation on an ABI stepOneplus system (Applied Biosystems, Waltham, MA, USA). The primer sets used are listed in the following Table 1:

Western blot: Western blot analysis was performed as previously described [20]. Briefly, sample concentrations were determined using the Bicinchoninic assay (BCA) according to manufacturer’s protocol and ran on a Mini-PROTEAN TGX Gels (Bio-rad). Primary antibodies against Bmi1 (1:1000, rabbit polyclonal, Abcam, Cambridge, United Kingdom, catalog ab38295), H3 (1:100, rabbit polyclonal, Cell Signaling, Danvers, MA, USA, catalog 9715S), Trimethylated H3 at Lysine 27 (1:500, rabbit polyclonal, Abcam, catalog ab195477), Acetylated Histone 3 (1:500, rabbit polyclonal, EMD Millipore, Burlington, MA, USA, catalog 06-599) and GAPDH (1:1000, mouse monoclonal, Millipore Sigma, Burlington MA, USA, catalog MAB374) antibodies (1:1000, LiCOR, Lincoln, NE, USA) were incubated for 1 h at room temperature, and visualized.

### 4.5. Proliferation Assays

Cell counting: the CBSCs transduced with the Bmi1 or scramble lentivirus were manually counted for 6 consecutive days. CyQUANT assay: To measure cellular proliferation, the CyQUANT Assay from ThermoFisher (C35011) was used following the manufacturer’s instructions on CBSCs that were treated with the Bmi1 or scramble lentivirus previously described. Fluorescence was measured on a spectrometer at excitation/emission of 508/527 nm.

MTT assay: the MTT Assay Kit from Abcam (ab211091) was used to measure cellular metabolism and proliferation following the manufacturer’s instructions on CBSCs that were treated with the Bmi1 or scramble lentivirus previously described.

### 4.6. Cell Death and Apoptosis Assays

Comet assay: to measure single-cell DNA damage, Bmi1 and scramble lentivirus CBSCs were plated on glass microscope slides with low melt agarose. The cells were lysed in a buffer solution described previously [41] for 1 h at 4C. The lysed cells on glass slides were placed in a gel electrophoresis box with 1 × TAE and run at 18 V for 1 h. The slides were stained with SYBR Safe and kept in the dark for viewing under a confocal microscope at 20 × objective. The images were analyzed using the OpenComet feature on Image J.

FACS sorting: the Bmi1 and scramble lentivirus CBSCs were exposed to 600 μM hydrogen peroxide for 12 h. The cells were stained and sorted on the LSR-II for Annexin V and DAPI to measure apoptosis and necrosis, respectively. The sort was analyzed using FlowJo software, Ashland, OR, USA.

### 4.7. Cell Cycle Assays

FACS sorting: the Bmi1 and scramble lentivirus CBSCs were exposed to serum starvation media for 24 h and then treated with 30 μmol/L H2O2 for 3 h the following day. Cell death was confirmed by visualizing the cells under a light microscope before collection. Cells were harvested and stained with Annexin-V (Life Technologies, Carlsbad, CA, USA.) and DAPI (EDM Millipore, Burlington, MA, USA) according to manufacturer’s protocol. Data were acquired with the BD fluorescence-activated cell sorting on the LSR-II and analyzed by Flow Jo or fluorescence activated cell sorting Diva software (BD Biosciences, Franklin Lakes, NJ, USA).

### 4.8. Histone Modification Assays

Mod Spec: RNA isolated from Bmi1 and scramble lentivirus CBSCs, as described previously, was sent to Northwestern for ModSpec analysis. Proteomics histone analyses were performed by the Northwestern Proteomics Core Facility.

### 4.9. Statistical Analysis

Analysis was performed by One-Way ANOVA (analysis of variance), followed by Tukey’s multiple comparison test using the GraphPad Prism software (GraphPad Inc., La Jolla, CA, USA). Unpaired T tests were used to compare the two groups. Statistical significance is shown with values of *p* < 0.05.

## Figures and Tables

**Figure 1 ijms-22-07813-f001:**
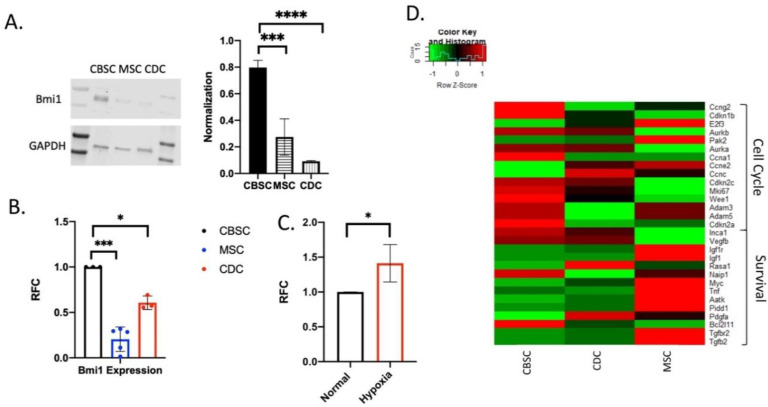
Bmi1 is overexpressed in CBSCs: (**A**) Western blot analysis of protein from cortical bone stem cells (CBSC), mesenchymal stem cell (MSC), and cardiac-derived stem cells (CDC) expressing Bmi1 and GAPDH (*p* values for CBSC vs. MSC = 0.026, *p* value CBSC vs. CDC = 0.011, *n* = 4); (**B**) PCR analysis of Bmi1 expression from the RNA from the three stem cell types in A (*p* values for CBCS vs. MSC *p* = 0.0003 and CBSC vs. CDC 0.031, *n* = 3); (**C**) PCR analysis of Bmi1 expression in CBSCs under normal versus hypoxic conditions (*p* value 0.042, *n* = 3); and (**D**) heat map comparison of cell cycle and survival genes between CBSCs, CDCs, and MSCs (*n* = 3). ). * *p* < 0.05, *** *p* < 0.001, **** *p* < 0.0001.

**Figure 2 ijms-22-07813-f002:**
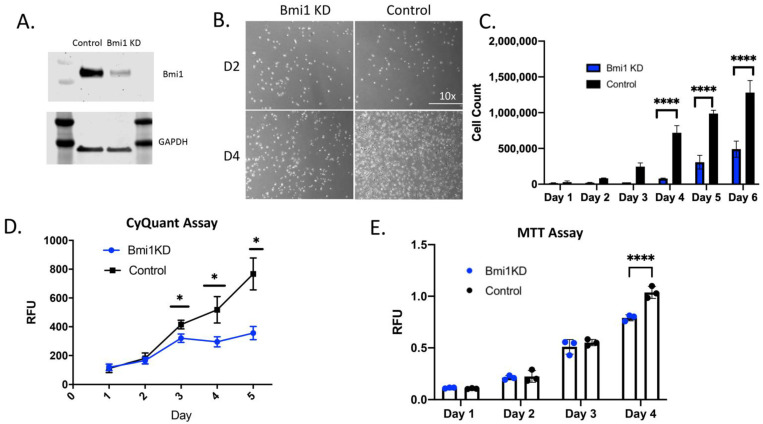
Bmi1 knockdown inhibits proliferation of CBSCs: (**A**) Western blot representative image of the control and the Bmi1 knockdown relative to GAPDH; (**B**) representative images of Bmi1 knockdown using shBmi1 lentivirus and the control using shScramble in CBSCs grown over 4 days; (**C**) cell count of the Bmi1 knockdown and control over 5 days (*p* value for Day 4 *p* < 0.0001 Day 5 *p* < 0.0001, *n* = 3); (**C**,**D**) Graph of the CyQuant assay comparing Bmi1 knockdown and control CBSC DNA content (*p* value for Day 3 0.0002, Day 4 < 0.0001, and Day 5 < 0.0001, *n* = 3); and (**E**) graph of MTT assay comparing Bmi1 knockdown to control CBSCs (*p* value for Day 4 < 0.0001, *n* = 3). * *p* < 0.05, **** *p* < 0.0001.

**Figure 3 ijms-22-07813-f003:**
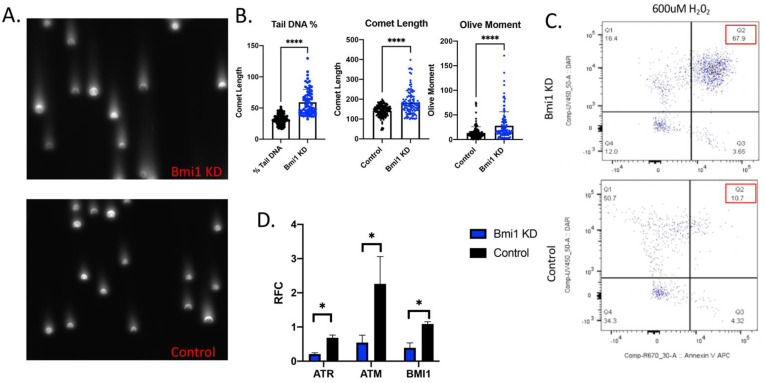
Bmi1 knockdown reduces survival and causes DNA damage in CBSCs: (**A**) representative images of the comet assay in the Bmi1KD CBSCs and the control CBSCs; (**B**) comet assay box plot analysis from the Bmi1 KD and the control CBSCs measuring Tail DNA Percent (*p* value < 0.0001), comet length (*p* value < 0.0001), and olive moment (*p* value < 0.0001 (*n* = 3); (**C**) flow cytometry gating strategy to determine apoptotic population. Dead cells were included in the analysis, and cells were gated on mCherry to ensure that only successfully infected cells were included in analysis. Percent of cells stained with AnnexinV positive and DAPI negative staining indicating apoptosis and necrosis (*n* = 3); and (**D**) the relative fold change (RFC) of PCR analysis using RNA from the Bmi1 KD and control CBSCs to assess DNA damage genes including ATR and ATM (ATR *p* value 0.001, ATM *p* value 0.022, Bmi1 *p* value 0.034, *n* = 3). * *p* < 0.05, **** *p* < 0.0001.

**Figure 4 ijms-22-07813-f004:**
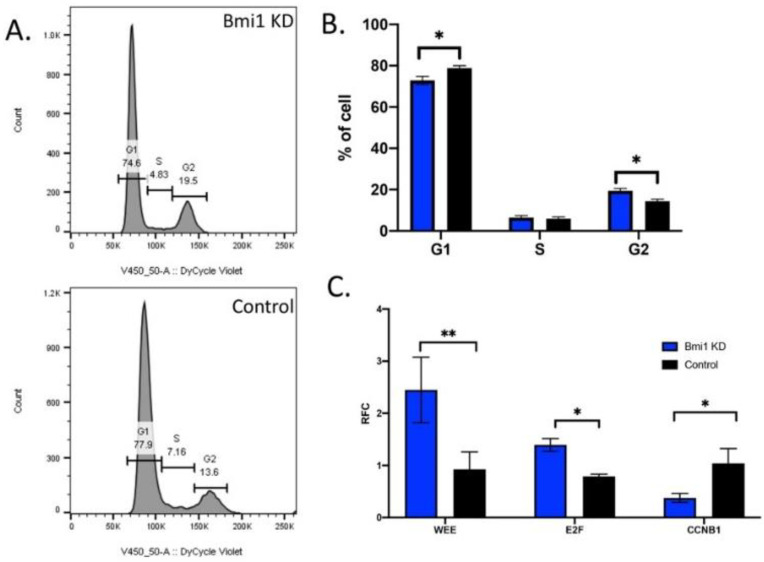
Bmi1 knockdown causes cell cycle arrest in CBSCs: (**A**) representative flow cytometry plots showing the cell cycle profile of serum-starved Bmi1 knockdown or control CBSCs; (**B**) graph showing the relative populations of G1, S, and G2 populations of CBSCs transduced with shBmi1 or shScramble (*p* value 0.007 for G1 and *p* value 0.003 for G2, *n* = 3); (**C**) the relative fold change (RFC) of PCR analysis shows that Bmi1 knockdown causes a decrease in cyclin B (*p* value 0.041), increase in WEE (*p* value 0.001), and an increase in E2F (*p* value 0.023, *n* = 3). * *p* < 0.05, ** *p* < 0.01.

**Figure 5 ijms-22-07813-f005:**
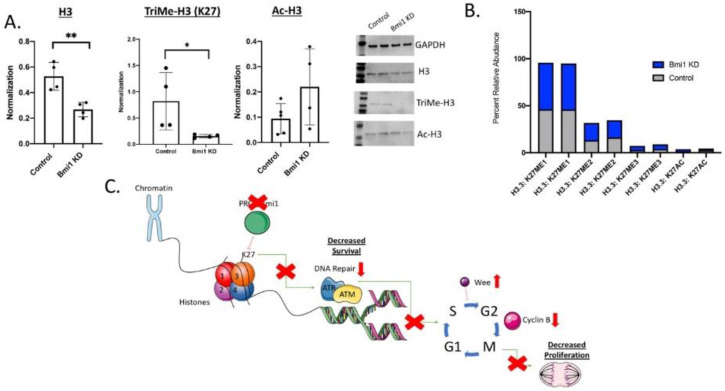
Bmi1 knockdown inhibits differentiation and modifies the epigenetic regulation in CBSCs: (**A**) Western blot analysis of histone 3 (*p* value 0.006), trimethylated histone 3 (*p* value 0.050), and acetylated histone 3 in CBSCs with and without Bmi1 using GAPDH for control (*n* = 4); (**B**) ModSpec data of Bmi1 KD versus control CBSCs for the percent of histone modification on H3K27 (*n* = 3); and (**C**) schematic representation of the mechanism describing the loss of Bmi1 in CBSCs. * *p* < 0.05, ** *p* < 0.01.

**Table 1 ijms-22-07813-t001:** Forward and reverse primer sequences.

Primer	Forward	Reverse
Bmi1	CCAGGGCTTTTCAAAAATGA	CGGGTGAGCTGCATAAAAAT
ATM	GCTAGTTCTGTGAGCGATGC	GCAGCTAAAGGACTCATGGC
ATR	GCGAATCATGACCCCTTTCC	CACATCATCGAAGCCTGCAA
CCNB1	AGCGAAGAGCTACAGGCAAG	TTCCACCTCTGGTTCACACA
E2F	CGAGGCCCTTGACTATCACT	AGGTCCCCAAAGTCACAGTC
WEE	AACCCCTTTACTCCGGATCC	TACGCTCTCTTTCTCCCACG
GAPDH	AGCGAGACCCCACTAACATC	TACGGCCAAATCCGTTCACA

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
