# Peer review of "Bmi1 Augments Proliferation and Survival of Cortical Bone-Derived Stem Cells after Injury through Novel Epigenetic Signaling via Histone 3 Regulation"

_ijms, 2021, doi:10.3390/ijms22157813_

Round 1
Reviewer 1 Report
Comments:
Overall, excellent study and presentation of novel discoveries.
Perhaps, add to the title something regarding "novel epigenetic signaling via Histone 3 regulation" as this is also a significant finding in this study.
Figure 5C deserves a separate Figure heading as it is a nice depiction of the signaling pathway. This should be discussed more in the Discussion, or at least referenced instead of Lines 150-153.
Line 209-210: I do not understand this line. What do you mean by role of Bmi1 role being modest?
Revisions:
Lines 12-14 to: Change to "Multiple stem cell types have been safely transferred into failing human hearts, but the overall clinical cardiovascular benefits have been modest."
Line 15: "have enhanced" to "enhance"
Line 26: "Concluding, that" to "In conclusion,"
Line 36: insert "often" before "irreversible"
Lines 75-82: Eliminate completely as this is a description of results which does not belong in the introduction section.
Lines 111-112: Eliminate completely as this is a summarizing statement, does not belong in results section.
Figure 4 caption, Line 181: Change "infected the" to "transduced with"
Line 195: Change "mechanism" to "mechanisms"
Line 223: "mark"??
Reviewer 2 Report
Lindsay et al submitted “Bmi-1 augments proliferation and survival of cortical bone derived stem cells after injury” for peer review. They concluded that CBSCs overexpress Bmi1 compared to other stem cells types, which enhanced the regulation of cellular proliferation, DNA damage, and cell cycle. This study was well designed. Results from this study supported their hypothesis. A revision is suggested.
- Please address the clinical limitations in this study.
- Please emphasize the clinical implications of this study.
- Please accurate to the third decimal place of p value.
- Please address RFC in each figure legends.
- Please quantify Fig3C(three repetitions).
- Please address how many repetitions for each assay in figure legends.
Round 2
Reviewer 2 Report
My questions had been well addressed.